# Legume Seed Protein Digestibility as Influenced by Traditional and Emerging Physical Processing Technologies

**DOI:** 10.3390/foods11152299

**Published:** 2022-08-02

**Authors:** Ikenna C. Ohanenye, Flora-Glad C. Ekezie, Roghayeh A. Sarteshnizi, Ruth T. Boachie, Chijioke U. Emenike, Xiaohong Sun, Ifeanyi D. Nwachukwu, Chibuike C. Udenigwe

**Affiliations:** 1School of Nutrition Sciences, Faculty of Health Sciences, University of Ottawa, Ottawa, ON K1H 8M5, Canada; iohaneny@uottawa.ca (I.C.O.); ekeziefloraglad@gmail.com (F.-G.C.E.); ramin063@uottawa.ca (R.A.S.); rboac063@uottawa.ca (R.T.B.); chijioke.emenike@dal.ca (C.U.E.); xiaohong.sun@dal.ca (X.S.); 2Department of Food Science and Technology, Faculty of Agriculture, Tarbiat Modares University, Tehran P.O. Box 14115-336, Iran; 3Department of Natural and Applied Sciences, Faculty of Science, Hezekiah University, Umudi, Nkwerre 471115, Nigeria; 4Department of Plant, Food and Environmental Sciences, Faculty of Agriculture, Dalhousie University, Truro, NS B2N 5E3, Canada; 5Center for Nutrition and Healthy Lifestyles, School of Public Health, Loma Linda University, Loma Linda, CA 92350, USA; 6Department of Chemistry and Biomolecular Sciences, Faculty of Science, University of Ottawa, Ottawa, ON K1N 6N5, Canada

**Keywords:** PDCAAS, DIAAS, in vitro digestibility, food processing, legume proteins, emerging food processing technology

## Abstract

The increased consumption of legume seeds as a strategy for enhancing food security, reducing malnutrition, and improving health outcomes on a global scale remains an ongoing subject of profound research interest. Legume seed proteins are rich in their dietary protein contents. However, coexisting with these proteins in the seed matrix are other components that inhibit protein digestibility. Thus, improving access to legume proteins often depends on the neutralisation of these inhibitors, which are collectively described as antinutrients or antinutritional factors. The determination of protein quality, which typically involves evaluating protein digestibility and essential amino acid content, is assessed using various methods, such as in vitro simulated gastrointestinal digestibility, protein digestibility-corrected amino acid score (IV-PDCAAS), and digestible indispensable amino acid score (DIAAS). Since most edible legumes are mainly available in their processed forms, an interrogation of these processing methods, which could be traditional (e.g., cooking, milling, extrusion, germination, and fermentation) or based on emerging technologies (e.g., high-pressure processing (HPP), ultrasound, irradiation, pulsed electric field (PEF), and microwave), is not only critical but also necessary given the capacity of processing methods to influence protein digestibility. Therefore, this timely and important review discusses how each of these processing methods affects legume seed digestibility, examines the potential for improvements, highlights the challenges posed by antinutritional factors, and suggests areas of focus for future research.

## 1. Introduction

Legume seeds are rich sources of dietary proteins and have emerged as potent tools for combating global malnutrition. Compared to animal protein sources, legume seeds are rich in dietary fibre which is beneficial to gut health, as well as health-promoting phytochemicals, and contain no cholesterol [1,2]. Furthermore, legume seed production is more sustainable and eco-friendly compared to animal production [3,4]. Therefore, legume proteins are considered healthier and greener alternatives to animal proteins. However, unlike animal proteins, which are highly digestible, legume seed proteins have reduced digestibility.

The digestibility of legume seed proteins is hindered by the protein structure, and, to a greater extent, by other components within the seed matrix such as trypsin inhibitors, phytates, tannins, and lectins, which are collectively known as antinutrients or antinutritional factors [5]. Trypsin inhibitors act on proteases, including peptidases, or, like phytates and tannins, form insoluble and indigestible complexes with proteins [6], which can lead to the alteration of protein structures, thereby limiting protease activity. Lectins act on the epithelial linings of the intestine by preventing the uptake of proteins, including digested proteins [7]. Therefore, the improvement of legume seed proteins greatly depends on the deactivation of these antinutritional factors or the release of proteins from the complexes formed with them [5]. Legume seeds are mainly consumed in their processed form and many of these processing methods have been reported to affect the digestibility of legume seed proteins. As summarised in Figure 1, processing methods are divided into traditional and emerging processing technologies. Traditional methods are those used in most households, such as cooking, baking, and milling, including the bioprocessing methods such as soaking, germination, or fermentation. Emerging processing technologies are continuously evolving with the development of new techniques, but for the purpose of this review, discussions are limited to high-pressure processing (HPP), ultrasound, irradiation, pulsed electric field (PEF), and microwave heating [8,9,10,11]. Unlike traditional processing methods, which are cheap, time-consuming, and require little or no complex technology, emerging technologies require certain levels of technological knowledge, are more rapid, and could be expensive.

Various methods are used to measure the digestibility of legume seeds, including in-vivo-, ex-vivo-, and in-vitro-simulated gastrointestinal digestion, with the last being the most commonly used method of assessment [12]. These methods have been criticised due to their over- or underestimation of protein digestibility, including the challenges of standardisation [13]. In response to some of these challenges, the protein-digestibility-corrected amino acid score (IV-PDCAAS) and digestible indispensable amino acid score (DIAAS) were developed. These two methods, in addition to measuring digestibility, also measure protein quality by the incorporation of the essential amino acids of the proteins [14]. Although both methods have been widely applauded for making readjustments to the perceived shortcomings of earlier methods, PDCAAS and DIAAS have their flaws.

By composition, most legume seeds contain up to 40% of their weight in proteins [15,16], but are low in fat (2–5%). Tissue-specific storage proteins are the most common proteins of legume seeds and their overall quality is influenced by their nutritional and functional properties [5,17]. Compared to animal proteins, plant proteins are low in certain essential amino acids, especially the limiting amino acids. Since animal proteins often contain considerable amounts of cholesterol and saturated fat, which are associated with various chronic diseases [18,19], plant-based proteins, including legume seed proteins, have increasingly become the preferred source of dietary protein for many consumers. Legume seed storage proteins, as with other dietary proteins, can be classified according to their physicochemical and functional properties such as solubility, gel formation, emulsification, and foam stabilisation as reported for proteins from pea, chickpea, lentil, lupine, and other bean types [5,20,21,22,23,24,25,26].

Globulins and albumins are the major legume seed storage proteins [20], when compared to glutelin and prolamin. Based on sedimentation coefficient (S20W), globulins are divided into two groups namely the 7S vicilin-type and the 11S legumin-type. Vicilin and legumin constitute 90% of the globulin proteins found in the protein body of *Vicia faba* and their molecular weight in different legumes as reported for the protein fractions of peanuts (*Arachis hypogea*), soybeans (*Glycine max*), the common bean (*Phaseolus vulgaris*), and broad beans (*Vicia faba*) [27,28,29,30]. The sulphur-containing amino acids (cysteine and methionine) and tryptophan are deficient in the 7S–11S globulins, which is a significant difference between globulins and albumins, as the latter is a functional protein as found in trypsin inhibitor and phytolectin, with higher sulphur-containing amino acids [30].

Thus, this review discusses how the structure and composition of legume seed proteins affect digestibility, the different methods used in the assessment of digestibility, the inhibition of protein digestibility by antinutritional factors, and the influence of traditional and emerging technologies on protein digestibility.

## 2. Assessment of Digestibility of Legume Seed Proteins

### 2.1. Methods Used in the Analyses of Legume Seed Protein Digestibility

Several methods have been used in the evaluation of protein digestibility, one of which is in vitro simulated gastrointestinal digestion. This method is divided into static and dynamic digestion tests, both of which involve the pH drop procedure by the addition of three different enzymes. In the pH drop technique, the pH is adjusted to 8, followed by the addition of enzyme mixtures containing trypsin, chymotrypsin, and protease from *Streptomyces griseus* or peptidase to facilitate digestion [31,32,33]. This pH adjustment [32,33] is typically performed in the first 10 min, and the following equation is used in the calculation of in vitro protein digestibility (IVPD):IVPD = 65.66 + 18.10 ∆pH10 min.

Despite its wide use, this method has been criticised for yielding digestibility results that are inconsistent with the actual proportion of digested proteins or cleaved peptide bonds [33]. Nevertheless, the high correlation of this technique with in vivo methods explains its use in many studies. An in vitro method has been recommended [34], in which the two-step digestibility process in the upper gastrointestinal digestive tract is imitated. In this method, after digestion of the protein for 1 h under the optimum conditions for pepsin, samples are digested with pancreatin at a similar concentration with pepsin for 2 h. As would be expected, these different approaches would return varying results, creating possibilities for discrepancies in digestibility; thus, creating the need for a standardised approach.

Given the need to obtain comparable data in different studies, a standardised static method was proposed by INFOGEST. To achieve this, more than two hundred scientists in the field of digestion from different countries worked together and proposed, for the first time, a three-step method that has been published [35], accompanied by comprehensive guidance on protein digestion in adults [36]. This three-step static model contains a detailed protocol for the oral phase, gastric phase, and intestinal phase in mimicry of in vitro digestion where the system for infants differs from that of adults in enzyme type, enzyme activity, and non-enzymatic factors. Although the pH of gastric phase is higher and oral phase is removed because of short resistance time [37], this model is used in the measurement of the digestibility of infant formula containing pea and faba bean proteins [38]. This method has made important contributions to the understanding of legume protein digestibility. For instance, it revealed that pigeon pea proteins such as convicilin, provicilin, vicilin, legumin A, and legumin A2 are not digested in the gastric phase but in the intestine [36]. Notwithstanding, the major criticism of the static method is its oversimplification of the digestion process.

Human digestion is dynamic and is a more complex process than is captured in these static models. Thus, the limitations of these static methods to predict in vivo digestion have been highlighted [39], such as consideration of a few simulated digestion parameters and different enzyme/substrate ratios, changes in pH, transformation through the gastrointestinal tract, and the removal of the digested materials under in vivo conditions [35]. For a better simulation of the in vivo digestion system, some dynamic (mono, di-, and multi-compartmental) models have been developed. These systems have incorporated some physiological factors such as the simulation of pH and temperature changes in the gastric and intestinal parts with the addition of bile, pepsin, and pancreatin, dialysis effect on absorption, gastric emptying time, mixing, and transit time. Collectively, these design systems have been considered to be at the highest simulation level [39]. Albeit, further improvements are needed since the inclusion of the roles of the gut microbiota, which are known to make significant contributions to protein and nutrient bioavailability [40,41], is still lacking.

Currently, in vivo digestibility measurement is based on the differences between nitrogen content of diet and faeces throughout feeding [42]. For instance, broiler chickens were used to compare the in vivo digestibility of different pea genotypes [43]. Moreover, the in vivo (rat model) method showed a good correlation (R^2^ = 0.75) with the in vitro (pH drop) digestibility method applied for five types of beans [42] and with red and green lentil (R^2^ = 0.89) [44]. Nevertheless, in vivo digestibility evaluation in humans is limited due to ethical, financial, and technical restrictions [37]; hence, the present reliance on different animal models.

As an alternative, the ex vivo method has been recommended for the evaluation of digestibility [45], where human gastrointestinal juice is used for digestibility analysis. In this method, after extracting juice from human bodies, pepsin activity of gastric juice and proteolytic activity of duodenal juice are measured. Samples are digested by gastric juice at pH 2.5 and then with duodenal juice with a pH of 7.5. This method was used to analyse the main proteins of pigeon pea isolate, which were digested and resulted in the discovery of peptide sequences with potential biological effects [46]. As such, the prospects of this method towards further understanding of the dynamics of digestibility are promising. However, wide exploration of this method in relation to the digestibility of other legume proteins is conspicuously lacking.

### 2.2. Legume Protein Digestibility

In a recent study, the in vitro protein digestibility of soybean and its byproduct (okara) was shown to be higher for okara (~68%) than soybean (~56%), a difference which was attributed to the higher phytic acid content of soybean compared to that of okara [47]. Similarly, using glutamic acid (Glu) equivalent of the INFOGEST in vitro digestibility method for the comparison of the digestibility of peanut, pigeon pea, and black bean, it was shown that the digestibility of pigeon pea protein was comparable to that of whey protein isolate (more than 0.02 mmol GLU equivalent) but was higher than that of peanut and black bean (at less than 0.01 mmol Glu equivalent) [36]. Furthermore, the evaluation of the protein digestibility of pigeon pea and chickpea protein (by the pepsin digestion method) showed a high digestibility value of up to 89% for both [48].

A study evaluating the possibility of incorporating faba and pea bean proteins into infant formula in comparison to rice and potato proteins, reported that the simulated in vitro infant protein digestibility was higher for pea and faba proteins, and was comparable to the digestibility of the milk reference protein [38]. The significance of this finding is further underlined by an earlier study [49], which reported that legume proteins’ digestibility was lower than the digestibility of animal proteins. Furthermore, the digestibility of legume proteins varies by type, with differential levels of digestibility among legumes in increasing fashion, reported to be soybean > lentil > chickpea > common bean, where the highest digestibility of soybean was attributed to its possession of the lowest β-sheet structure [49,50].

A reverse correlation was reported between the digestibility of native and processed legume proteins and their β-sheet content in protein structure [50]. The majority of proteins with reduced digestibility are usually of the 7s vicilin and 11s legumin, as identified [51] for chickpea proteins under in vitro digestion. The main structure in these proteins is β-barrel structure, and they belong to the seed storage protein group known to possess nutraceutical and allergenic effects [50]. Pigeon pea vicilin digestibility was found to be higher in the gastric phase compared to that of other legumes [36], perhaps due to the high solubility of 7s vicilin [52].

In a different study, the digestibility of legumes was compared based on the cell wall effect where the physical or enzymatic removal of the cell walls increased legume digestion by up to 50% [53]. This suggests that physiological barriers such as cell wall influence protein digestibility. For instance, a comparative study on the digestibility of mung bean, pea, chickpea, and red kidney bean revealed that mung bean and red kidney bean had the highest and lowest digestible proteins, respectively, with the degradable fragile nature of mung bean cell wall being credited for its high digestibility [54]. Furthermore, Zahir, Fogliano, & Capuano [55], indicated that cell wall permeability contributed to the improved digestibility of soybean, which occurred as a result of enhanced trypsin diffusion following processing. Thus, it is thought that the apparent disruption of cell wall integrity following processing improves legume protein digestibility [56].

Previous methods of protein digestibility assessment have often been criticised for over- or underestimation of digestibility, resulting in the need for further improvement of digestibility analysis, as seen with new methods such as in vitro protein-digestibility-corrected amino acid score (IV-PDCAAS) and digestible indispensable amino acid score (DIAAS). The PDCAAS method compares the amount of the essential amino acids in a food sample to a scoring pattern in order to determine its most limiting amino acid (amino acid score), but has been critiqued for having a maximum achievable score of 1.0, and thus it does not accurately rank proteins of a higher quality, whose scoring values are greater than 1.0. Unlike PDCAAS samples, which are taken from faeces, ileal samples are used for DIAAS, thus offering a more accurate estimate of digestibility and bioavailability, since amino acids are absorbed in the ileum. DIAAS scores >1.0, unlike those of PDCAAS, are also not rounded down to 1, thus permitting a more accurate estimation of digestibility. IV-PDCAAS was used to evaluate the pH drop method and it showed a high score for soybean and chickpea (desi and kabuli) flour (72–82%), while red lentil showed a lower quality score of 43% [57]. Similarly, DIAAS was used to evaluate protein quality and digestibility, which, although analogous in pattern to red and green lentil, had lower values compared to IV-PDCAAS scores [57]. In another study, PDCAAS obtained by an in vivo rat bioassay was higher for processed red lentil than the green lentil [44]. The lower digestibility of green lentil has been previously reported [58].

A comparison of the protein quality of green and yellow split peas (*Pisum sativum*) by DIAAS and PDCAAS reported a higher score for green split pea compared to yellow split peas [58]. DIAAS evaluation for Canadian pulses showed that pinto bean and navy bean had higher scores compared to black beans [59]. Based on the reference protein used for PDCAAS and DIAAS calculation, DIAAS value was found to fall between the PDCAAS in vitro and in vivo values [57,58]. Taken together, these studies showed that the evaluation of legume protein digestibility makes the comparison of results difficult. This challenge has been eased by the INFOGEST proposed standard method, which has produced more comparable results. Nevertheless, in relation to the comparability of INFOGEST method and in vivo digestibility of legumes, additional studies are warranted.

## 3. Inhibition of Digestibility

### 3.1. Antinutrients

Antinutritional factors are compounds which are known to inhibit the digestibility and thus impair the nutritional quality of various foods, including legume food proteins. These antinutritional factors or antinutrients, which may be heat-stable or heat-labile, include enzymes (trypsin, chymotrypsin, and α-amylase) inhibitors, polyphenolics such as tannins, phytates, lectins (or phytohaemagglutinins), saponins, vicine, convicine, gossypol, uricogenic nucleobases, metal chelators, and cyanogenic glycosides [60,61,62,63]. Antinutrients may occur naturally in foods (e.g., trypsin inhibitors in soybean) or may only appear in foods following processing (e.g., Maillard reaction products in soybean-based feed). Since they are proteins, trypsin inhibitors can be inactivated by various heat-processing methods such as extrusion, micronisation, flaking, and autoclaving, among others [60], although certain trypsin inhibitor activities in chickpea albumin have been reported to be resistant to heat [62]. Kunitz and Bowman–Birk inhibitors are two main forms of trypsin inhibitors found in abundant amounts in soybean that are not readily inactivated by heat processing as a result of the presence of disulphide bonds [63].

Disulphide bonds, when present in feed, could severely impair the activity of proteolytic enzymes in the gastrointestinal tract of animals [64]. In addition to enzyme inhibitors, tannins, which are present in considerable quantities in beans and peas, are known to complex with and impair the digestibility of proteins [62], and a number of processing methods such as autoclaving, soaking, and dehulling have been found to be effective for reducing tannin levels [60,61]. It has been suggested that the high tannin content of beans, peas, and some cereal crops could further contribute to the protein malnutrition of people living in certain regions of the world, especially where those foods form part of the diet staple [60]. This tannin-related low digestibility and poor feed utilisation efficiency have been reported in cattle and sheep, but not in goats, which are resistant to tannins [63]. Similarly, phytates were implicated in the low digestibility of kidney beans when compared to chickpea, faba bean, and peas; this was demonstrated to be consistent with the high phytic acid content of kidney beans [65]. Legumes are also abundant in lectins and glycoproteins, which can easily adhere to erythrocytes, resulting in agglutination [63]. The inhibition of digestibility by lectins could be seen in their capacity to limit nutrient absorption by binding to intestinal epithelial cells and to impair the growth of laboratory animals [63]. Hence, this results in the suggestion that lectins might be inhibitors of protein uptake as opposed to inhibitors of protein digestibility [5].

### 3.2. Mechanisms of Inhibition

Trypsin inhibitors are amongst the most studied antinutrients, given their potential for wide applications in health and nutrition. Most trypsin inhibitors act by binding to substrates (in this case trypsin) in a competitive inhibition mode [66]. Since they bind at the active site of the protease (trypsin), trypsin inhibitors prevent trypsin from binding to proteins and catalysing proteolytic reactions. Phytate, another antinutritional factor which is common in legumes, is known to complex with proteins at both acidic and basic pH, and by so doing, alter protein conformation, hinder enzymatic action, and inhibit digestibility [67]. Since calcium ions are essential for the activity of trypsin and α-amylase, phytate chelation of Ca^2+^ also contributes to the inhibition of digestibility by the antinutrients [67]. Lastly, polyphenolic antinutrients with bulky structures such as tannins are known to inhibit digestibility by disrupting, destabilising, or loosening enzyme structure/conformation [68]. It is believed that the majority of tannins inhibit enzymatic action by allosteric regulation rather than the inhibition of specific single active sites [68].

### 3.3. Food Structure and Matrix Effect

Mature legume seeds are intact, metabolically dormant, and cannot be broken down by human digestive proteases. Hence, this leads to the report that postharvest storage conditions result in “hard-to-cook” grains due to interactions between phytate, the mineral cations, and pectin, which increase seed toughness, thereby reducing their susceptibility to softening during cooking [69]. Hard-to-cook quality has been attributed to high accumulations of β-sheets and α-helix, including 31, 30, and 27 kDa polypeptides, as found in kidney beans when compared to easy-to-cook legumes [70]. These qualities were implicated in the inhibition of trypsin activities, leading to the low protein digestibility associated with hard-to-cook legume seeds [70]. Thus, the role of food structure on protein digestibility is of significance.

The matrix of a food protein, as with structure, is known to influence its digestibility, and ultimately, its bioavailability and absorption [71]. It is thought that the differential degree of protein crosslinking and aggregation in primary legume proteins, such as globulins and albumins from pea, fosters different protein matrices which could affect protein digestibility differently, such that the disulphide-bond-rich albumin compact structure is not as amenable to proteases as that of the globulin protein [72]. Like other food proteins, legume proteins do not exist in a vacuum but instead co-occur with and form complex matrices with other components of foods, such as phytochemicals, metal ions, dietary fibre, phytic acid, lipids, and carbohydrates, all of which could shield them from and/or impair the digestive action of proteases, and thus limit their digestibility. For instance, it is believed that the matrices formed by the presence of polyphenols, phytic acid, and lectins in the globular proteins of the common bean, *Phaseolus vulgaris* L., contribute to their poor digestibility [73]. In fact, various studies comparing the digestibility of these bean globular proteins (also called phaseolin or vicilin) have reported widely varying results, which can be attributed to the different microstructure of the globulins used. As discussed by Rovalino-Córdova et al. [73], digestibility values ranging from <61% to >80% were observed for various common bean protein samples (flour, fractions, and beans that differed in their globulin type) with varying sample microstructure, and which thus differed in the nature of the interaction between their different food matrices and the digestive enzymes.

In addition, the binding to and/or physical entrapment of legume proteins in cellular structures has been shown to inhibit their digestibility. The digestibility of various legume proteins could be dictated by the specific interaction of the protein cell wall with enzymes in the gastrointestinal tract since cell walls are made up of complex heterogeneous structures such as cellulose, hemicellulose, and pectin, whose exact composition and organisation vary according to the plant [74]. In a recent study, soybean cell wall, which forms part of the cellular matrix of the legume’s cotyledons, was found to inhibit its digestibility [55]. Using soybean samples with diverse matrices such as uncooked cotyledons with intact cell wall, cooked cotyledons with intact cells, protein samples extracted from cooked and uncooked cotyledons, as well as boiled-then-mashed and mashed-then-boiled samples, the authors studied the effect of food matrix interactions on in vitro legume protein digestibility. Extracted soybean proteins without the structural barrier of a tightly packed cytoplasm were found to be significantly more digestible (41% higher percentage degree of hydrolysis) than the samples largely confined within the legume’s intact cells, and thus they existed in a more or less “crowded cellular environment” where their interaction with other components of the food matrix apparently contributed to their lower digestibility [55]. It was concluded that the intact cell wall acts as a barrier, which could delay or inhibit the proteolytic action of digestive enzymes [55].

In addition to the barrier function of the intact cell wall, it was also hypothesised that cellular integrity could inhibit digestibility by presenting a compact intracellular environment which would further restrict the flow of proteases or impair enzyme motility within the cytoplasm [55,75]. Further evidence of the influence of physical entrapment, structural organisation, and matrix compactness of legume proteins on digestibility was provided in the study by Rovalino-Córdova et al. [73], involving red kidney bean samples with intact cotyledon cells (ICC), mechanically damaged cells (MDC, obtained from the physical disruption of ICC), and bean flour solution, (BFS). Data from the study show that, in general, the BFS sample (which was obtained following the aqueous suspension of red kidney bean flour after soaking, dehulling, drying, milling, and sieving) had the highest degree of hydrolysis for both the simulated gastric and intestinal phases of the in vitro digestion, followed by the MDC sample [73]. Clearly, since the proteins in the BFS sample were free from the kind of physical boundaries and molecular confinement engendered by the tightly packed cytoplasm and compact cell wall of the other samples, they enjoyed relatively less restrained access to the proteases used in the study. Thus, the compact structure of the intracellular cytoplasmic matrix of the legume proteins, which contains—among other things—protein bodies and starch granules, may have together combined to place a structural constraint on the digestibility of the ICC and MDC samples by inhibiting the access of digestive enzymes to these substrates [73].

The idea that inter-individual differences in oral comminution could represent a significant determinant of inter-individual differences in nutrient bioavailability was recently put forward in a perspective article [71]. Apart from cell wall integrity, the particle size of legume food proteins, whether determined by a rudimentary processing method such as mastication [71] or by a more sophisticated method such as milling [33], could influence its matrix, and thus its digestibility. The role of oral processing in determining food matrix properties is crucial given its position as the first step in food digestion and the inter-individual differences in mastication, and the consequent implications for digestibility and nutrient bioavailability [71]. It is conceivable that a more thorough comminution would have a greater impact on bolus matrix properties and would thus exert a more substantial effect on particle size and cell wall integrity of legume food proteins than a more superficial dental compression [71]. Interestingly, differential glycaemic index values following the consumption of rice meals have been attributed to such inter-individual differences in mastication [76], and could be an important factor in legume protein nutrition.

## 4. Effects of Traditional Processing Technologies on Protein Structure and Digestibility

Various food processing methods are known to improve the digestibility of legume seed proteins [5]. Indeed, majority of legume seeds used for human food consumption are in their processed form. The most common practices in many households are the traditional/bioprocessing methods including thermal treatments such as boiling, microwaving, autoclaving [77], baking, and frying. As detailed in Table 1, these methods can disrupt the compact cell and protein structure and the entire matrix or modify their functional properties, thus improving their digestibility, bioavailability, nutritional value, and functionality. Nonetheless, these processing methods are tailored to their applications.

### 4.1. Bioprocessing: Imbibition, Germination, and Fermentation

Seeds when soaked imbibe water, which reactivates intracellular metabolic activities culminating in germination [5]. Seed germination is accompanied by the remobilisation of the storage proteins such as legumin and vicilin, causing significant changes in the profile of proteins and other seed components, including the reported enhancement of antioxidant properties, yield of α-glucosidase, and DPP-IV inhibitory peptides [94,95,96]. Protein content seems to progressively increase from soaking to germination [56]. For instance, Dipnaik and Bathere [97], found a sequential increase in the protein content of legume seeds soaked for 12 h and then sprouted (from 30 ± 1.07% to 32 ± 0.57%, then to 40 ± 0.57% and from 32 ± 1.8% to 40 ± 1.0%, then to 48 ± 0.57% for cowpea and chickpea control, soaked and germinated seeds, respectively). The authors reported that the measured transaminase activity, as an indicator of new protein synthesis, increased in accordance with the increases in seed protein contents [97]. Both soaking and germination have been reported to increase legume seed protein digestibility [5,56,98]; perhaps this is due to the increase in the hydrolytic enzyme activities [99] or the removal of antinutrients [100]. In addition to soaking and germination, fermentation is another traditional processing method widely employed in legume seed processing. During fermentation, the microbial metabolism involves some proteolytic activities which degrade the seed cell wall, causing a change in the seed microstructure and alteration of the protein profile [95]. Thus, fermentation has been reported to increase seed protein digestibility [100], which could be due to the degradation of the cell wall [54,55,75] or disruption of the secondary structure of the proteins [70], both of which the authors reported as having improved legume seed protein digestibility. Regardless, these studies showed that bioprocessing improved legume seed protein digestibility.

### 4.2. Physical Treatment: Milling, Thermal Treatment, and Extrusion

The milling of legume seeds, which includes detaching the seed coat from endosperm and disrupting the cell wall, is used in advanced applications. Milling into flour reduces particle size and increases the surface area for enzyme contact and activity, and, as such, has been reported to improve the digestibility of legume seeds [101]. Particle size reduction disrupts the cell wall integrity; thus, the reported improvement of digestibility attributed to milling could also be due to the alteration of cell wall that enhances legume seed protein digestibility [75]. Besides milling, legume seeds can also undergo processing by thermal treatment, which has also been reported to improve protein digestibility [102], with variations in digestibility depending on the type of legume and cooking time [72].

In one study, thermal treatment was found to denature native protein structure and modify the structure of protease inhibitors along with legume storage proteins while also causing protein aggregation, thus making the proteins more susceptible to digestive proteases during unfolding of the protein [103]. In another study, Ma et al. [104] evaluated the effect of boiling (90 °C for 20 min) or roasting (80 °C for 1 min) on trypsin inhibitor activity, the functionality, and the microstructure of desi chickpea, kabuli chickpea, red lentil, green lentil, and yellow pea flours. The authors reported that both boiling and roasting significantly reduced trypsin inhibitor activity in all flours, although the extent of reduction varied by legume type. Thermal treatment enhances structural changes, which are further improved by wet heating as gelatinisation and crosslinkages occur between proteins and starch. However, non-polar bonds and intramolecular hydrogen bonds are significantly affected by thermal treatment, which changes the native conformation. In addition, most comparable studies treated whole seeds prior to milling, thus making the evidence of gelatinisation and crosslinkages less observable, as was the case with boiled legume flour samples, where boiling directly lowered protein solubility [104,105]. Whereas the untreated samples were highly soluble at pH 7 (between 53.7% and 61.04%), at pH 5, the solubility was reduced to 10% for the treated and untreated samples. Understandably, pH 5 is close to the isoelectric point of 4.6 and the reduced solubility could also be due to disulphide-sulfhydryl interchange reactions [104,105].

Extrusion is a form of severe thermal treatment that can influence the amino acids content and improve the quality and nutritional values of legume seed proteins such as the reported increase in the phenylalanine content of kidney beans [72,77]. In contrast, the valine, phenylalanine, and lysine contents decreased in peas extruded at 129 °C and the tryptophan content was reduced after extrusion at 142 °C, thus reducing the PDCAAS values [106,107]. Extrusion at high temperature can cause the condensation of the carbonyl group of reducing carbohydrates and the free amino groups of amino acids (or proteins/peptides) in a reversible reaction known as early Maillard reaction. However, the advanced Maillard reaction is irreversible when the condensation has proceeded to cyclisation and the conversion of the aldose formed in the early stages to ketone occurs in an irreversible reaction. Furthermore, extrusion was reported to reduce the tannin content and the trypsin inhibitor activity, with concomitant enhancement of in vitro digestibility [107,108,109]. Collectively, the impact of the processing method on protein quality is highly influenced by the legume type and treatment. For instance, cooked green pea had lower PDCAAS and DIAAS compared to samples subjected to extrusion and baking, in contrast to cooked split yellow peas, which returned higher PDCAAS and DIAAS [58]. However, it is not clear whether the result was influenced by using whole seeds in the boiling experiment as opposed to the flour used for extrusion and baking.

### 4.3. Combination of Bioprocessing and Physical Treatment

Previously, a lower level of free amino groups after heating was reported, and a combination of heating and soaking reduced the nitrogen content [110]. The authors attributed this decrease in free amino group to Maillard reaction between the carbonyls of either fructose or glucose and the amino groups present. However, the legume seeds were milled after and not before soaking, cooking, or autoclaving. As such, the factors involved in protein solubility may have contributed to the reported reductions, since processing methods that influence these factors are known to alter the interaction with proteases [111]. Moreover, the soaking of *V. unguiculata* in 0.05 g/100 mL sodium bicarbonate solution for 12 h increased the in vitro protein digestibility to 75.28 ± 0.36% compared to the control and soaking in water at 71.28 ± 0.12% and 73.16 ± 0.28%, respectively. Although these studies have shown that the combination of physical and bioprocessing methods improved legume seed protein digestibility, the extent of improvement is influenced by the chosen combinations. For instance, germination and cooking were reported to have slightly enhanced digestibility, but improvement was more pronounced when germination for at least 72 h was combined with autoclaving [112].

## 5. Emerging Processing Technologies on Protein Structure and Digestibility

### 5.1. High-Pressure Processing (HPP)

Advances in technology and changes in consumers’ preference towards the direction of natural products have triggered a transition in the food industry from traditional methods involving the application of heat to novel food processing technologies. High-pressure processing (HPP) is one of the new technologies at the forefront of this trend, with huge commercial interest due to its benefits such as significant decline in processing times, while maintaining food safety. HPP technology can also exert positive effects on food ingredients and components such as protein quality, by modifying the conformational and structural arrangement of proteins, which influences their various protein functionalities in food systems or in the human body [88]. The extent of these changes varies with factors such as the level of pressure applied, temperature, duration of treatment, and the food matrix. HPP can inactivate or diminish several antinutritional factors (ANFs) and metabolites, including the flatulence-causing oligosaccharides, trypsin inhibitors, tannins, and phytic acid, which form indigestible complexes that are resistant to even high quantities of digestive enzymes, thereby improving the digestibility of legume foods [88,113].

The application of HPP technology towards improving digestibility and the nutritional value of legumes has been demonstrated in many studies, where the harmonised conclusion was that HPP induces unfolding of protein structure, thereby facilitating the accessibility of proteolytic enzymes. HPP interrupts the balance of several intramolecular interactions such as hydrophobic and van der Waals interactions, with minimal impact on covalent bonds [87]. An earlier finding, which formed the basic concepts used in recent works, indicates that the amount of flatulence-causing sugars in isolated proteins and flour fractions of selected legumes (i.e., bean, lupine, and pea), was modified by the disruption of protein-saccharide, accompanied by a decline in gas production, measured in vitro [114]. In addition, a combination of two different pressurisation instances at different time intervals has also been evaluated in whole split peas and white beans with reference to specific ANFs, i.e., trypsin inhibitor, oligosaccharides, and phytic acids, and the corresponding effect of in vitro digestibility (IVPD) [87]. Regardless of the duration of treatment, the authors found a decrease in these ANFs when treated at a pressure level of 600 MPa [87]. This outcome reflected positively on the IVPD of the whole legumes, which increased for both split peas and white beans by 4.3% and 8.7%, respectively [87]. Other works indicated that it is also possible to achieve a significant reduction in the soaking time of whole legume seeds with a simultaneous enhancement of their digestibility using high pressure [6,87].

As stated earlier, exposure of legume protein to pressurisation at optimum conditions (i.e., temperature, treatment duration, etc.) facilitates the accessibility of peptides to sites of enzymatic digestion depending on the type of protease used and the pH of the medium. The rationale for this phenomenon can also be explained from a macroscopic point of view, as noted for protein structures of pressure-treated protein isolates from red kidney bean [115]. Depending on the applied pressure, the protein structures were visualised to be largely irregular in shape, with a good number of pores developed on the surfaces accompanied by a reduction in particle size [115]. Protein dimers were possibly dissociated into monomers, causing the reported decrease in particle size. The opening of the tissue structure and reduction of small pores following the application of high-pressure aid in enzymatic digestion. Since protein function is dependent on its structural integrity, the series of structural modifications due to denaturation, fragmentation, and aggregation of proteins [115,116], induced by HPP technology, makes it a valuable tool for improving the functionality and digestibility of legume proteins. However, to achieve desirable outcomes, optimal conditions with respect to pressure level, time, and duration of treatment should be considered.

### 5.2. Ultrasound

The use of ultrasonic waves is a promising and practical approach for improving the functionality of food ingredients in the food and beverage industry. The application of ultrasound in aqueous medium leads to the formation of cavitation bubbles which collapse violently, resulting in a spike in temperatures and pressures that yields turbulence in the cavitation alongside high shear energy. Cavitation fragments food matrices and facilitates the release of protein, including ANFs such as saponins, isoflavones, and polyphenols from plant cells, and is thus beneficial for improving protein quality [117]. During protein extraction, sonication can also reduce particle size by ~10-fold as a result of an amplified surface area [117]. There are several advantages in using high-intensity ultrasound (HIU) process over traditional thermal methods. They include short processing time, improved functionality of food ingredients, low cost and energy consumption as well as maximum retention of nutritional value, among others [118].

An increase in the digestibility of legume protein is generally associated with a decrease in levels of ANFs in legumes following treatment [93,119]. With ultrasound, virtually all the saponins were accessible in ethanol extracts of different edible legumes [120], accompanied by a considerable level of inactivation of soybean trypsin inhibitors as reported in other studies [10,121]. So far, only a few studies have focused on the impact of ultrasound on ANFs in legumes, but for the purpose of this review, we undertake a discussion of the possible mechanism through which ultrasound could influence the physicochemical properties of proteins, especially in relation to digestibility.

Firstly, sonication can modify the molecular configuration of proteins by cleaving hydrogen bonds, and hydrophobic interactions lead to a loss in structural integrity. These changes can alter protein function and could therefore influence protein digestibility. In addition, HIU is said to induce changes in free sulfhydryl content. Specifically, the high pressure and sheer force of cavitation from sonication can transform the S-S linkage to free -SH groups, exposed to the surface of proteins. Hence, ultrasonication was shown to lead to an increase in the free –SH groups of pea protein isolated by power ultrasound for 5 min [122]. It is also possible that the peroxides (H_2_O_2_) present in the gas bubbles during sonication triggered oxidation of free –SH groups and formed S-S bonds. The disruption of non-covalent interactions by sonication transformed the structure of the pea protein isolates PPI from large, intertwined aggregates into smaller dense entities with uniform size distribution [123,124]. This cumulative structural modification may have been responsible for the 2–4% higher IVPD values recorded for soluble protein concentrates of selected legumes compared to their raw counterparts [125]. However, a more recent report indicated that ultrasound as a physical treatment showed no effect on the IVPD of pigeon pea flour [11]. Aside from differences in processing conditions and grain type, the precise explanation for the disparity is not clear since limited works are currently available for comparison.

Furthermore, protein microstructure was demonstrated to play a crucial role in the protein digestibility of plant food, with protein aggregates highly susceptible to proteolysis [126]. In one study, compared to whole cells of unsonicated samples, a good number of microfractures were observed for sonicated soy flakes and flour sample, with debris on the surface suggesting a breakdown of cell and layer formation [127]. The cracks/fissures observed for sonicated chicken pea flour facilitate the liberation of proteins and other components [127], which can enhance the action of digestive enzymes. Additionally, it was microscopically illustrated that at higher intensities and times, smaller structures of protein particles can be visualised in sonicated pea protein isolates (at 105–110 W/cm^2^ for 40 min) [128] and soy protein isolates (at 45–48 W/cm^2^ for 30 min) [129]. Partial protein unfolding and weaker intermolecular interactions exerted by strong cavitational forces may be responsible. These findings emphasise the relevance of microstructure and structural unfolding of protein in legume protein digestibility following ultrasonication, but no detailed direct study was found in the literature.

### 5.3. Irradiation

Irradiation is a processing method in which food is exposed to ionising radiation in a contained environment under process-controlled conditions. In addition to antimicrobial and enzyme inactivation effects of radiation, this processing method is capable of inactivating or removing some unfavourable compounds in legumes that impair digestibility [80]. For example, Bamidele and Akanbi [84] studied the effect of irradiation on the IVPD of pigeon pea flour and found a dose-dependent increase in the IVPD of the raw flour at an irradiation dose of 20 kGy. The increase can be ascribed to the breakdown of proteins into fractions susceptible to enzymatic action. Moreover, the observed decrease in antinutrients (e.g., phytates, lectins, protease inhibitors, oxalates, etc.) present in pigeon pea might have contributed to the increase in IVPD. These results also correspond to an earlier report on the impact of gamma irradiation at different dosages on the nutritional quality of powdered velvet bean seed, where excluding an irradiation dosage at 2.5 kGy, all other treatments showed a significant decline in the phytic acid compositions, and complete degradation was reached at 15 and 30 kGy [130]. A positive result (i.e., significant reduction of phytic acid and increase in IVPD) was also reported for carioca beans exposed to γ-radiation using ^60^Co as source [79], in the presence of hydration. The free radicals generated by irradiation can disrupt the bond between phytate and other components, leading to the chemical degradation of phytate to phosphates and inositol or the splitting of the phytate ring. In addition, tannins content is also affected by irradiation, as observed for fava and common beans [131,132]. Although a clear pathway that led to this outcome on tannins remains uncertain, the process may vary with the type of grain, composition, and/or metabolic activity.

Irradiation can also reduce the cooking time of grains by 50% following previous hydration [133]. This irradiation-hydration process also softens the grain and improves the extraction and degradation of these antinutritional compounds, e.g., in irradiated faba bean cultivars where an increase in IVPD values was also confirmed [132]. The authors noted that phytates can release the bounded sodium and potassium, which can replace the calcium and magnesium from the seed coat’s pectate, leading to bean softening and a decrease in cooking time. In general, gamma irradiation may cleave non-covalent bonds that maintain protein structure, via generated free radicals that attack protein molecules, leading to a loss of conformational integrity that exposes additional peptide bonds and improves proteolysis and protein digestibility. Other studies have also documented the positive influence of irradiation towards increasing the protein digestibility of whole soybeans, common beans, canola seeds etc. [79,134,135]. Even so, indigestible phytohaemagglutinins (PHA), a type of plant lectin that causes severe intestinal damage which disrupts digestion, have also been successfully inactivated by up to 50% in purified and seed forms [136]. In general, the process of irradiating food depends on several factors such as radiation dose, food matrix, oxygen exposure, temperature, and storage time. It is important that these factors are controlled for desirable results where applicable.

### 5.4. Pulsed Electric Field (PEF)

Pulsed electric field (PEF) is an emerging food processing technology known for its effectiveness against microorganisms in different food products. In PEF processing, food is inserted between two electrodes and is exposed to short but strong pulses of high voltages (up to 80 kV/cm), with high efficiency. Regarding the impact of PEF technology on legume protein digestibility, a paucity of information is currently available in the literature. However, Li et al. [137] showed a decline in the concentration of soybean trypsin inhibitors as the intensity of pulses applied increased, though the time of treatment had a less obvious effect. This study suggests that further enquiry into the corresponding response in IVPD may show an increased trend. Other investigations revealed that PEF can alter the protein profile of non-plant foods such as beef and induce structural changes with a favourable effect on in vitro digestion kinetics by affecting the protein interactions [138]. In addition, structural changes caused by interference with the individual polypeptide chains, intermolecular interactions, and disulphide bonds trigger partial unfolding of proteins, which enables legume proteins to be easily digestible [139]. The heterogeneous charge distribution lying on the protein backbone causes the protein molecule to slowly stretch or deform during PEF processing [140]. Further, the ohmic heat generated during the process can induce protein denaturation and aggregation [120,128]. Presently, the exact effect of PEF processing on the IVPD of legume proteins is still unclear; hence investigations focused on the basic mechanisms will be invaluable, especially for developing new protein structures and legume foods with improved digestibility and overall nutritional value.

### 5.5. Microwave Heating

Microwave cooking is a thermal treatment in which food is exposed to microwave radiation. Compared to traditional methods, microwave processing reduces the time required to inactivate microorganisms in food products at safe levels. This cooking method has been fairly studied for the reduction of ANFs responsible for limiting the digestibility of legume proteins. For example, microwave heating was found to be considerably effective in reducing protease inhibitors in various legume seeds [72]. Similar results were achieved using microwave treatment at 2450 MHz for 10 min in eliminating trypsin inhibitors in velvet beans [141]. Comparatively, the efficacy of dielectric heating is higher for inactivation of ANFs compared to using high temperature and pressure treatments [90].

Further, the underlying mechanism through which microwave enhances the IVPD of legume protein digestibility is not fully clear. A study recently proposed an explanation for the significant increase in IVPD (54.4 ± 2.5% to 71.6 ± 4.2%) of pigeon pea flour, from a microscopic point of view [11]. The authors found that SEM images showed the formation of starch-protein cluster components with a characteristic larger size, pores, and increased surface roughness in the microwave-treated flour. During microwave processing, the increase in temperature caused starch gelatinisation and swelling of starch granules, leading to collapse of the granular structure [142]. The formation of surfaces and internal pockets in the microwave-treated flour facilitates the accessibility of enzymes to the proteins [143]. To a large extent, the modification of secondary structures also affects the digestibility of proteins [144]. For instance, a negative linear correlation between β-sheet/turn and IVPD of soy proteins was reported [145]. An increase in random coils and decline in both α-helix and β-sheet for microwave-treated sample confirmed the formation of unfolded structures and better structural flexibility, which can enhance protein digestibility [144]. These results were supported by the declining trends observed for trypsin inhibitor activity and tannin contents, with a corresponding increase in IVPD values [146].

Microwave as a pre-treatment prior to the soaking of legumes is also beneficial for protein functionality and digestibility. This was demonstrated in a work by [93], where soaked/microwave chicken pea seeds had the most pronounced elimination of tannins accompanied by significant reduction in phytate. Compared to the control, protein solubility in microwave-treated samples was lowered significantly, while IVPD significantly improved. Other reported studies showing an increase in IVPD caused by microwave treatment include black soybeans [90], chickpea [93], and soymilk [10]. One of the advantages of using microwave irradiation to improve the overall nutritional quality of legumes is that the processing time essential for attaining safe levels of ANFs inactivation is lower, compared to traditional processing technologies [91].

Lastly, other advanced heating methods such as radiofrequency (RF) treatment, ohmic heating, infra-red treatment, and non-thermal technologies such as pulsed light and cold plasma technology may potentially be effective for reducing ANFs in foods, accompanied by increase in protein digestibility. For instance, RF treatment effectively decreased ANFs in black soybean [90]. Whereas infrared treatment at 1342 W for 15 min led to a considerable decline in trypsin inhibitors in soaked soybean seeds [146]. These studies further demonstrate the potential of novel non-thermal methods in promoting the protein digestibility and overall quality of leguminous foods. However, an important factor to be considered is ensuring maximum retention of other nutritional components during processing.

## 6. Conclusions and Future Directions

Factors contributing to the impairment of legume seed protein digestibility are multifaceted. Although antinutrients are the most studied contributors to legume seed protein digestibility, other factors such as food structure and matrix, legume type, and processing methods are also vital. Legume seed digestibility can be enhanced using various food processing methods, whether they are traditional or emerging technology methods. Traditional methods of processing are time-consuming when compared to the emerging technologies, but the cost and technical knowhow demands of emerging technologies may be a hindrance to their popularity. Although food processing methods enhance legume seed protein digestibility, the extent of improvement is determined by the type of treatment, the length of treatment, and the legume type. For instance, some processing methods, while decreasing the level of antinutrients in legumes, could simultaneously reduce or adversely affect other legume components. Therefore, future studies should focus on identifying what processing methods are suitable for what legume types for optimal and effective incorporation into food production and design.

## Figures and Tables

**Figure 1 foods-11-02299-f001:**
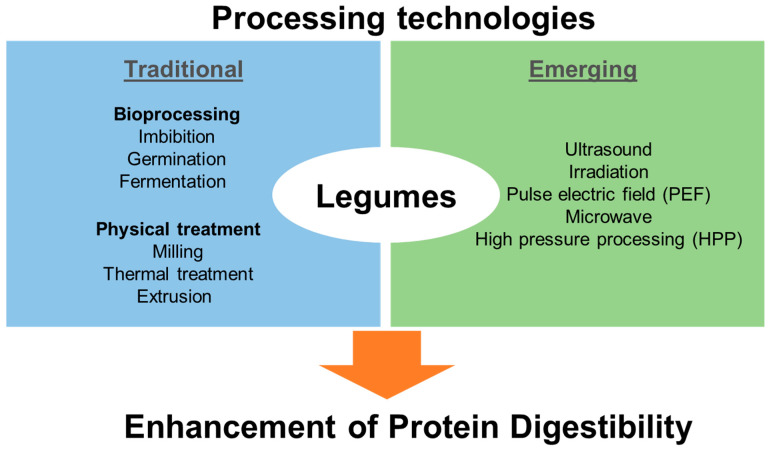
Overview of the major traditional and emerging food processing technologies applied for enhancement of the digestibility of legume proteins.

**Table 1 foods-11-02299-t001:** Influence of emerging food processing technologies on the protein digestibility of selected legume seeds.

Technology	Legume Type	Treatment Conditions	IVPD Outcome	Process Advantages	Process Disadvantages	Reference
Irradiation	Soybean	4.8, 9.2, 15.3, and 21.2 kGy	+	Increases isoflavone, phenol, and anthocyanin content	May induce vitamin C reduction.	[78]
	Carioca beans	10 kGy	+			[79]
	Faba beans	0.5 and 1.0 kGy	+			[80]
	Moth beans	2, 5, 10, 15, and 25 kGy	+			[81]
	Cow pea	2, 5, 10, 15, and 25 kGy	+			[82]
	Sesame seed	0.5 and 1.0 kGy	+			[83]
	Pigeon pea	20 kGy	+			[84]
Ultrasound	Soybean	20 kHz and amplitude (20–40%), 10–20 min	+	Selective method	Energy-demanding	[85]
	Soybean	25 kHz, 400 W, 1–16 min	+			[10]
	Fava bean	20 kHz, 750 W	−			[86]
High-pressure processing	Split peas White beans	600 MPa, 60 °C, different time intervals	+	- Enhances shelf-life - Retains and improves organoleptic characteristics	It should be used in combination with other methods to achieve high effectiveness	[87]
	Peas	600 MPa, 25–28 °C, 5 min	+			[88]
	Red bean	600 MPa, 25 °C, 5 min	+			[89]
	Black soybean	600 MPa, 60 °C, 30 min	+			[90]
Microwave heating	Soybean	2450 MHz, 2–10 min 70–100 °C,	+	- Very high efficiency - Process can be fully standardised. - Enhances aromatic amino acids. - Reduces raffinose, stachyose, and verbascose	- Energy-demanding - Reduces solubility - May affect sulphur-containing amino acids - Decrease in minerals (Na, Ca, and Mg) - Decrease in B vitamins	[10]
	Soybean	2450 MHz, 1000 Watts	+			[91]
	Yellow field pea seeds	1200 W, 25 min	+			[92]
	Pigeon pea	Cook mode, 3 min	+			[11]
	Chickpea	2450 MHz, 15 min	+			[93]
Radiofrequency heating	Black soybean	2450 MHz, 30 min	No effect	- Energy-saving - Efficient - Low nutritional losses - Reduced cooking time	- Reduced power density	[90]

Note: + increase, − decrease.

## Data Availability

No new data were created or analyzed in this study. Data sharing is not applicable to this article.

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
