# Peer review of "Legume Seed Protein Digestibility as Influenced by Traditional and Emerging Physical Processing Technologies"

_foods, 2022, doi:10.3390/foods11152299_

Round 1
Reviewer 1 Report
The review is interesting, however, some points need to be improved:
Line 21. Specify components that inhibit protein digestibility.
In the abstract, there is no information about the main findings.
Line 89. This reference is not reported in refences section
Section 2.1 PDCAAS and DIAAS methods were not explained.
Line 252. Comparable results, why authors conclude that are comparable results
Line 414. Correct the reference format
Line 543. What are optimum conditions
Line 629-630. How much reduction has been found?
A table for values comparison would be helpful
Correct format for reference 26, 45,
Author Response
The review is interesting, however, some points need to be improved:
*The authors are grateful for this comment.
Line 21. Specify components that inhibit protein digestibility.
*The next sentence immediately identifies these inhibiting components as "antinutrients or antinutritional factors" which are then described in details subsequently in the manuscript.
In the abstract, there is no information about the main findings.
*This is a review paper and therefore does not contain any major findings. However, the objective of the work is clearly set out in the last sentence as follows: "Therefore, this timely and important review discusses how each of these processing methods affects legume seed digestibility, examines the potential for improvements, highlights the challenges posed by antinutritional factors and suggests areas of focus for future research."
Line 89. This reference is not reported in refences section
*That reference has now been replaced by two more recent and more relevant references, which have also been included in the reference list. Thank you.
Section 2.1 PDCAAS and DIAAS methods were not explained.
*Thank you. Explanations have been provided for both methods.
Line 252. Comparable results, why authors conclude that are comparable results
*The authors' conclusion regarding the comparability of INFOGEST results is supported by many works including these two: (A). Brodkorb, A., Egger, L., Alminger, M., Alvito, P., Assunção, R., Ballance, S., ... & Recio, I. (2019). INFOGEST static in vitro simulation of gastrointestinal food digestion. Nature protocols, 14(4), 991-1014. and (B) Egger, L., Ménard, O., Delgado-Andrade, C., Alvito, P., Assunção, R., Balance, S., ... & Portmann, R. (2016). The harmonized INFOGEST in vitro digestion method: From knowledge to action. Food Research International, 88, 217-225.
Line 414. Correct the reference format
*Done. Thank you.
Line 543. What are optimum conditions
*Details regarding the said optimum conditions have been added to the manuscript. Thank you.
Line 629-630. How much reduction has been found?
*The cited work does not provide any more detail than the stated "significant reduction"
A table for values comparison would be helpful
*Thank you for the suggestion. A table has been included.
Correct format for reference 26, 45,
*Thank you. Done.
Reviewer 2 Report
This article describes methods of assessment of the digestibility of the protein, intrinsic and processing-related factors affecting the digestibility of legume seed proteins. Traditional processing and relatively novel technologies including, high-pressure processing, ultrasound, irradiation, pulsed electric field and microwave treatments are reviewed. The topic of this study is interesting and follows current trends in food science and technology (especially issues concerning novel technologies), thus the information included in the manuscript can be valuable for potential readers. The introduction provides the essential background of the work. In general, modes of information presentation are clear, the work is well organized and comprehensively described. The most important information is provided in a concise form. Conclusions summarise the most relevant findings and outline further perspectives. The references are appropriate and adequate, as well as in most cases come from recent years. In my opinion, despite fact that work is generally well written, the main drawback is the lack of visual „attention grabbers” like figures, schemes, tables or even graphical abstract, which, although not obligatory, but it would increase the quality of the manuscript. Therefore, I recommended minor revision.
Author Response
The authors are grateful for the comments and suggestions of Reviewer 2. A table and a set of figures have now been included to further enrich this work.
Reviewer 3 Report
The manuscript has reviewed the effects of traditional and emerging technologies on the digestibility of legume seed proteins. The review is appropriately written and contains useful information. The work can be accepted after addressing the following remarks:
The format of references must be corrected.
The manuscript is very long without any tables or figures. Please present some data in tables or figures.
Line 251: add space “This challenge”
Author Response
The manuscript has reviewed the effects of traditional and emerging technologies on the digestibility of legume seed proteins. The review is appropriately written and contains useful information. The work can be accepted after addressing the following remarks:
*The authors are grateful for Reviewer 3's helpful comments.
The format of references must be corrected.
* The format of the references has been modified.
The manuscript is very long without any tables or figures. Please present some data in tables or figures.
A table and a set of figures are now included in the manuscript. Thank you.
Line 251: add space “This challenge”
* Done. Thank you.